# 'I can't cope with multiple inputs': a qualitative study of the lived experience of 'brain fog' after COVID-19

Caitriona Callan,[1] Emma Ladds ![ORCID] ,[1] Laiba Husain,[1] Kyle Pattinson,[2] Trisha Greenhalgh ![ORCID] [1]

¹Nuffield Department of Primary Care Health Sciences, University of Oxford, Oxford, UK
²Nuffield Department of Clinical Neurosciences, University of Oxford, Oxford, UK

**Correspondence to**
Dr Emma Ladds;
e.ladds@nhs.net

## ABSTRACT

**Objective** To explore the lived experience of 'brain fog'—the wide variety of neurocognitive symptoms that can follow COVID-19.

**Design and setting** A UK-wide longitudinal qualitative study comprising online focus groups with email follow-up.

**Method** 50 participants were recruited from a previous qualitative study of the lived experience of long COVID-19 (n=23) and online support groups for people with persistent neurocognitive symptoms following COVID-19 (n=27). In remotely held focus groups, participants were invited to describe their neurocognitive symptoms and comment on others' accounts. Individuals were followed up by email 4–6 months later. Data were audiotaped, transcribed, anonymised and coded in NVIVO. They were analysed by an interdisciplinary team with expertise in general practice, clinical neuroscience, the sociology of chronic illness and service delivery, and checked by people with lived experience of brain fog.

**Results** Of the 50 participants, 42 were female and 32 white British. Most had never been hospitalised for COVID-19. Qualitative analysis revealed the following themes: mixed views on the appropriateness of the term 'brain fog'; rich descriptions of the experience of neurocognitive symptoms (especially executive function, attention, memory and language), accounts of how the illness fluctuated—and progressed over time; the profound psychosocial impact of the condition on relationships, personal and professional identity; self-perceptions of guilt, shame and stigma; strategies used for self-management; challenges accessing and navigating the healthcare system; and participants' search for physical mechanisms to explain their symptoms.

**Conclusion** These qualitative findings complement research into the epidemiology and mechanisms of neurocognitive symptoms after COVID-19. Services for such patients should include: an ongoing therapeutic relationship with a clinician who engages with their experience of neurocognitive symptoms in its personal, social and occupational context as well as specialist services that include provision for neurocognitive symptoms, are accessible, easily navigable, comprehensive and interdisciplinary.

## Strengths and limitations of study

► To the best of our knowledge, this is the largest and most in-depth qualitative study of the lived experience of brain fog in survivors of COVID-19.

► The research team was interdisciplinary and interprofessional, and included consultation with two patient experts by experience suffering from ongoing, improving brain fog, who helped with data interpretation and peer review.

► Oversampling from men and non-white ethnic groups allowed partial correction of an initially skewed sample.

► The sample was drawn entirely from the UK.

► Residual skews in the samples, particularly regarding minority ethnic groups and occupational classes and the digitally excluded, limited our ability to capture the full range of experiences.

a UK sample suggest 1 in 10 people self-report ongoing, otherwise unexplained symptoms 12 weeks after infection.[1] Over half experience a reduced functionality for everyday activities and many remain unable to work weeks after infection.[2] The growing frequency of chronic and/or disabling illness related to COVID-19 has rendered their health needs, and associated clinical and occupational guidelines, policy priorities.[3–5]

Long COVID-19, a 'patient-made' term,[6] embraces the formally defined ongoing symptomatic COVID-19 syndrome (symptoms persisting between 4 weeks and 12 weeks) and post-COVID-19 syndrome (symptoms beyond 12 weeks).[5] In this paper, we use 'long COVID-19' to refer to the lived patient experience and 'post-COVID-19 syndrome' to refer to the medically diagnosed condition. It is highly heterogeneous with sufferers reporting a range of fluctuating symptoms, among which fatigue, breathlessness, chest pain, post-exertional malaise, autonomic nervous system disruption and cognitive dysfunction[4 7–9] are common. The pathophysiology remains unclear; however,

## BACKGROUND

It is now well-established that symptoms can occur beyond acute COVID-19. Results from

persistent viraemia,[10] relapse or reinfection[11] inflammatory and immune reactions,[12 13] deconditioning[14] and psychological factors[15 16] have been proposed as contributors. It is likely that causative pathways are multifactorial.[17]

Analysis of a quarter of a million COVID-19 survivors' health records revealed widespread neurological and psychiatric presentations with around one-third persistently affected over the following 6 months.[18] Around one-quarter experienced disturbed mood, and a fraction developed serious problems such as psychosis. Other neurological problems have included cerebrovascular events, encephalitis, dementia and disorders of peripheral nerves, nerve roots or plexuses.[18] Surveys and focus groups of online, non-hospitalised long COVID-19 patients have identified subjective and/or objectively measured impairments in attentional processing, short-term memory and executive function, alongside a befuddled state termed 'brain fog' by many patients.[4 7 9 19] A few studies have explored correlations between subjective cognitive dysfunction and neuropsychological testing deficits with mixed findings.[20–22]

In this paper, we use patients' own descriptions of their symptoms (using their term 'brain fog') and, when appropriate, the US National Cancer Institute definition of 'neurocognitive symptoms' to describe subjective problems 'to do with the ability to think and reason, [including] the ability to concentrate, remember things, process information, learn, speak, and understand'.[23] Possible proposed biological factors include direct neuro-invasion,[24] viral persistence and chronic inflammation,[25] neuronal injury or toxicity and glial activation,[24 26] microvascular injury,[27] activation of autoimmune mechanisms[28] and Lewy body production,[29] while imaging demonstrates loss of grey matter in patients with COVID-19 in key brain regions.[30]

The functional impact of such neurocognitive symptoms is often profound, affecting individuals' abilities to work and perform daily activities,[4 9] increasing healthcare contacts,[31] impeding decision-making, communication and social relationships. UK clinical guidelines suggest that clinical psychology and psychiatry specialists should be a part of the multidisciplinary team conducting post-COVID rehabilitation,[5] but these are contested and inconsistently implemented. Developments in treatment approaches, service pathways and occupational supports require better understanding of underlying causal and contributory factors as well as the lived experience of sufferers. 'Brain fog' has been highlighted in previous research as a particularly impactful aspect of long COVID-19, which sufferers are keen to have further explored.[4 32 33]

In this study, we sought to answer three key questions: (a) 'what neurocognitive symptoms are experienced by adults with long COVID-19?'; (b) 'what is the impact of these symptoms?' and (c) 'how do individuals deal with them?'. We also sought to explore whether our understanding of cognitive processes/perceptions and the COVID-19 could inform potential causative explanations.

## METHODS

### Study design and governance

This study extended a previous qualitative study of 114 people with self-defined long COVID-19.[4 32] Original recruitment took place between May 2020 and September 2020 from support groups on Facebook, a social media call (Twitter) and snowballing. To correct skew, men and minority ethnic groups were oversampled. In October 2020, partly prompted by participants' desire to further explore brain fog, the original sample was emailed for focus group volunteers—23 agreed. Twenty seven additional participants were then recruited from an online support group dedicated to long COVID-19's neurocognitive effects. The dataset for this study thus consisted of data from the original study and focus groups from the new sample of 50. In line with ethics committee recommendations and infection control measures, email or verbal consent was obtained.[4]

Five focus groups of 60–90-minute duration were held via Zoom in October and November 2020 with 10–14 participants. Each group had two facilitators (el and LH—female researchers experienced in qualitative research with qualifications in general practice and public health) who also took contemporaneous notes. Participants were invited to tell the story of their neurocognitive symptoms, with conversational prompts to maintain the narrative and elicit the impact on an individual's life and any interaction between neurocognitive and other perceptually 'physical' symptoms.[34] We encouraged the sharing of stories to identify issues important to the patient, emotional touchpoints in their illness journeys and promote interaction between participants.[35]

### Data management and analysis

Focus groups were videotaped with consent, transcribed in full, de-identified and entered onto NVIVO software V.12 alongside contemporaneous notes. Additional material from the original dataset was included. Sections of text were initially coded by CC (a female researcher qualified in psychology and medicine and training in qualitative methodology) into six categories: naming the phenomenon; lived experience of symptoms; interaction of neurocognitive and other symptoms; impact of symptoms; self-management; and experiences navigating healthcare services. These were informed by, but not limited to, the theoretical framework discussed below.

An interim synthesis was produced from early transcripts and progressively refined using the constant comparative method by CC and el.[36] Finally, to add descriptive depth, clarify discrepancies or ambiguities within the data and track progression of symptoms, we sent each participant a follow-up email 4–6 months later (10–12 months after their acute illness). We asked how their symptoms were progressing and to describe their current neurocognitive symptoms. Twenty participants responded and this data were integrated into, and refined, our final interpretation. While saturation did not determine sample size, thematic saturation was reached.[37]

## Theoretical framework

Our analysis was informed by three theoretical lenses. First, we considered the symptom burden of long COVID-19 from a neuroscience perspective. For many, long COVID-19 symptoms are poorly explained by objective medical tests. Although this may relate to undiagnosed peripheral pathophysiology, there is an increasing appreciation that unexplained symptoms also relate to the brain's perceptual processes.[38 39] The brain has no direct access to the body or outside world and must make sense of noisy incoming sensory signals. Current theories propose signals are deciphered by referring to an internally held model of perception.[38 39] This can be influenced by multiple factors, including mood, previous experiences and conscious or unconscious beliefs. Thus, symptoms can be generated, exacerbated or perpetuated independently of a cause 'in the body'.[38–40] In the case of COVID-19, SARS-CoV-2 is neuroinvasive, and thus additionally may directly disrupt these perceptual processes.[38]

Second, sociological theories of chronic illness, including May's burden of illness theory,[41] biographical perspectives on chronic illness[42–44] and the sociological notion of stigma.[45] Third, emotional touchpoints of powerful feelings such as anger, fear or hope[46]—particularly in participants' experiences of healthcare, which may be interpreted using theories of good professional practice,[47] the therapeutic relationship[48] and continuity of care.[49]

## Patient involvement statement

The study was planned, undertaken, analysed and written in collaboration with participants suffering from long COVID-19. All were invited to a webinar presentation sharing key findings and quotes, provided with a recording and copy of the presentation, and invited to correct errors or misinterpretations, which largely reflected a desire to ensure the severity of their symptoms and their impacts were appropriately represented. Although the recovery status of all participants is unknown, 13 of the 20 follow-up respondents had ongoing but improving brain fog 10–12 months after initial infection. Furthermore, two clinically qualified people still suffering from long COVID-19 reviewed a near-final draft of this paper, which was modified in response.

## RESULTS

### Description of dataset

Details of participants are shown in table 1. Despite our efforts to balance for gender and ethnicity, the final sample was skewed to 42 of 50 (84%) female and 36 (72%) white. By comparison, long COVID-19 support

| Table 1 | Participant characteristics | | | |
|---|---|---|---|---|
| | Participants recruited from previous long COVID study | Participants recruited from neuro COVID support groups | Total brain fog focus group participants | Responders to email follow-up post-focus groups |
| | 23 | 27 | 50 | 20 |
| Gender | | | | |
| Female | 15 | 26 | 42 | 17 |
| Male | 8 | 1 | 8 | 3 |
| Age | | | | |
| Median | 48 | 36 | 43 | 43 |
| Range | 31–74 | 29–68 | 29–74 | 31–74 |
| Ethnicity | | | | |
| White British | 16 | 14 | 30 | 11 |
| White other | 3 | 3 | 6 | 1 |
| Black | 1 | 1 | 2 | 0 |
| Asian | 3 | 2 | 5 | 1 |
| Mixed | 0 | 0 | 0 | 0 |
| Non-response | 0 | 7 | 7 | 7 |
| Occupation | | | | |
| Healthcare professional | 8 | 8 | 16 | 5 |
| Non-healthcare professional | 13 | 11 | 24 | 9 |
| Non-response | 2 | 8 | 10 | 6 |
| Hospitalised at any point due to COVID-19 | | | | |
| Yes | 0 | 4 | 4 | |
| No | 9 | 8 | 17 | 4 |
| Non-response | 14 | 15 | 29 | 16 |

groups are up to 86% female[9] and the UK population is 80%–85% white British.[50] The five focus groups, chat transcripts, follow-up email communications and participant webinar discussion produced over 1000 pages of transcripts and notes. The six emergent coding themes are discussed in more detail below with illustrative quotes in table 2 and definitions of neurocognitive processes/functions in box 1.

1. Naming the phenomenon.
2. Neurocognitive symptoms and their natural history.
3. Neurocognitive symptoms in the context of other long COVID-19 symptoms.
4. Psychosocial impact: guilt, shame and stigma.
5. Hypothesising mechanisms to inform self-management.
6. Navigating healthcare.

### Naming the phenomenon

Participants varied in their attitudes toward the patient-made term 'brain fog'.[9] Some found it useful as an accessible and well-known shorthand to disclose their wide-ranging cognitive difficulties to others, but others felt the term lacked specificity or did not convey the severity of their symptoms (quote 1). Alternative terms preferred by some participants included 'clinical or profound brain dysfunction', 'neurocognitive fatigue' or 'brain impairment', although all participants used the term 'brain fog' in group discussions.

### Neurocognitive symptoms and their natural history

This study focused on patients' lived experiences with no objective examination. However, their descriptions often related to specific domains of cognitive function—particularly, executive function, attention, memory and language, with most describing difficulties across all of these domains. Participants described problems with planning, decision-making, flexibility and working memory, which concorded with executive function cognitive processes (quote 2), while impairments in complex attention included difficulties with selective, sustained attention, divided attention and processing speed (quote 3), and long-term memory impairments were experienced with free recall, cued recall and procedural memory (quote 4). Language deficits varied between individuals, including difficulties with word finding and fluency, syntax, reading comprehension and writing (quote 5).

The longitudinal email follow-up allowed us to explore some aspects of the condition's natural course. Most respondents reported emergence of neurocognitive symptoms 1–4 months after their initial illness, and 13/20 felt they had improving brain fog at time of follow-up. Neurocognitive symptoms tended to fluctuate diurnally and over weeks to months, typically, but not invariably, showing gradual long-term improvement (quote 6). The tiring and unpredictable nature of the symptoms were destabilising and debilitating and were reported similarly among all participants.

### Neurocognitive symptoms in the context of other long COVID-19 symptoms

Participants described having distinct experiences of 'neurocognitive' compared with 'physical' symptoms. The latter were generally presented as somatic manifestations, often familiar from other conditions, such as physical fatigue, tachycardia or breathlessness. Despite this distinction, there was a recognition that both 'physical' and 'neurocognitive' symptoms were often associated or interacting. Many highlighted the fatiguability of their neurocognitive or physical symptoms from either mental or physical effort (quote 7).

### Psychosocial impact: guilt, shame and stigma

Participants described profound psychological, occupational and social impacts. Several had been unable to return to work at their previous level or at all. Participants who had returned to work described adopting reduced hours or adapted roles, often associated with anxiety about potential risks associated with mistakes in cognitively demanding or high-responsibility roles (quote 3), self-doubt about their abilities, loss of self-worth and altered identities (quote 8).

Participants reported how their symptoms induced strong emotional responses in themselves and others. Guilt and shame were particularly evident, often relating to difficulties in returning to work, their previous level of function or a lack of understanding from others (quotes 9 and 10). Particularly, troubling was physically invisible deficit, such as difficulties with language or memory. Participants also described instances of conflict arising from their impaired cognition (quote 12).

### Hypotheses to inform self-management

Participants frequently attempted to make sense of their symptoms and communicate the severity and legitimacy of their suffering through analogous referral to disorders with accepted mechanisms such as stroke, concussion or dementia (quotes 14 and 16). Although of those who had been 'investigated' many were 'normal', participants were keen to hypothesise about biological explanations for their symptoms with some also mentioning psychological contributors to their experience. Some reported various self-management strategies based on hypothetical mechanisms such as dietary adaptations (quote 6), food supplements or complementary therapies, which were met with variable success.

Many had developed coping strategies to deal with their neurocognitive symptoms, centred around self-expectation management and rest prioritisation, resulting in self-negotiations and activity trade-offs, which were frustrating and psychologically draining (quote 11). Moreover, conveying their reduced and variable cognitive function to family, friends or colleagues was a significant challenge and some developed innovative communication strategies (quote 12).

**Table 2** Participant quotes

| Identifier | Source | Quote |
|---|---|---|
| 1 | Participant 10, FG4 | 'Does anyone ever refer to it as neurocognitive fatigue? In a way I don't like brain fog as it's too vague, too loose of a term, so want something more technical. Though I don't think neurocognitive fatigue encompass the word finding difficulties, so it's not ideal either.' |
| 2 | Participant 7, FG1 | 'One of the things I've realised is how many things I do in my normal day - I'm not talking about work, just in a normal day - that are cognitive that I (didn't previously)think of as being cognitive. So a supermarket, the amount of sensory information, and just staring at a row of things looking for the food that you want, remembering where things are in the aisles and planning your trip so that you don't have to walk backwards and forwards around the shop, that surprised me. (…) Not just can I walk around the supermarket, it's planning, it's getting there, it's choosing stuff, all of that is actually really difficult.' |
| 3 | Participant 5, FG1 | 'I can't cope with multiple inputs, like if I'm trying to reply to a message on my phone and one of my boys starts speaking to me or there's something else happening as well that just really fries my brain. I mean I used to be the kind of person that, like all women, multi-tasking was a superpower. I was able to, do lots and lots of things, you know I'm [a doctor]; I would have one patient I'd be hearing lots about another patient coming I'd be remembering I'd be doing something else I'd be juggling lots and lots of things and now I can't keep multiple plates spinning I absolutely can't. I've got to focus on just one thing or I make massive mistakes and it's like I forget my intentions all the time.' |
| 4 | Participant 10, FG3 | 'I can ask somebody a question and then I'll ask the exact same question 2 min after and not remember I've asked them, I can't remember significant things that have happened in the past either.' |
| 5 | Participant 8, FG2 | '(It's difficult)to comprehend and take in written information and read it. I had a form sent to me at work and I just felt, 'I can't do this at the moment' and put it to one side and hoped to come back to it because it's just been too difficult.' |
| 6 | Participant 3, FG5, in email response to follow-up | 'I'm probably about 90% better. I'm struggling to put in full days at work and still need a great deal of rest and sleep. My brain fog is greatly improved, although I'm making mistakes at work and have been forgetful and sometimes confused with large amounts of new information. I feel like my head is clear now. When you did the group interview I felt like I was drugged up all of the time. Now it's far and few days between that I feel that way. I think the brain fog lasted around 8 months. I strongly believe that my improvements are diet related and have been following a low histamine diet since October.' |
| 7 | Participant 2, FG1 | 'Sometimes I feel as though if I exert myself like cognitively then my Long COVID-19 symptoms sort of exacerbate like shortness of breath, chest tightness. But like earlier on I think that it was the other way round (…) it seemed to be that if I exert myself physically-this means going for a 5 min walk on flat-then I get confused, I can't remember stuff, so it's like I find it really hard to unpick which way round it is.' |
| 8 | Participant 11, FG3 | 'Seven months plus in I don't know whether I'm gonna get my brain back(…)I'm really, really fearful for the future or whether I'm going to be able to get back to what I want to do and that's like your identity and yourself and who I am as a person is, you know, a big part of me is being a (allied health professional) and if I can't, if I've lost that, I've lost a huge part of me.' |
| 9 | Participant 9, FG4 | 'I found myself restating and reiterating many times professionally where I'm at now in terms of cognitive ability and there's only so many times you can do that before I feel like I'm becoming that person, you know and it's a lot easier to do that in the house but I think professionally it's been really hard.' |
| 10 | Participant 5, FG4 | 'a few times that I've been out and had an in-depth conversation with somebody that hasn't managed to get used to how I am, they've sort of said to me 'you're going round in circles in your conversation' or 'you're not making a lot of sense', when I hadn't quite recognised how repetitive I was being until somebody said it back to me. But even so those same people … can't seem to cut me any slack for it, or can't seem to understand how difficult it is, do you know what I mean? [There] just doesn't seem to be the understanding there and I can understand that because it would be beyond my comprehension as well if I hadn't lived it.' |
| 11 | Participant 5, FG2 | 'For me it's been going from working at 110% pace to not being able to get out of bed, not being able to work to not see people, to have to cancel plans, the impact on my life has been a massive transition and getting my head around that has been huge. I'm accepting now that I need to take the time off to get better and although that's really difficult and it's meant letting lots of people down, and there's been a complete change in my life, I've managed to get to that place.' |
| 12 | Participant 7, FG4 | 'Me and my husband have got a traffic light system now, so green's fine, he can just talk business at me, amber is like can you just keep 'what's the weather'-like kind of conversation, and then red is just stop, I need to just rest, stop all the sensory input coming in. And that seems to be working quite well now, so literally I've got to say amber or red and it's that thing when you're so tired that you can't even articulate that you're so tired and explain. So that really has helped us and I think might stop quite a lot of rows.' |
| 13 | Participant 5, FG3 | 'I find it extraordinary difficult-doctors, GP's that I spoke to, I just couldn't seem to put it across at all, they would just sort of think 'well why are you worrying, of course you're ill, you're not thinking properly, it will pass'. I couldn't seem to get across the enormity of how much it's affected me and how many different struggles there'd been. And I think part of that is because my communication has actually been impaired from it.' |
| 14 | Participant 8, FG1 | 'I have to say it was when my GP said 'yes, we recognise what you've got as Long COVID-19 and we're treating it like concussion at the moment until we know more about it, and we will recommend you rest and maybe try these drugs', I mean, I almost broke down it was the acknowledgement of the issue. [It] takes away so much of the stress because, we're all [thinking], you know, 'is this really happening, is this just me malingering or do I really have this thing'. And so that was that was a key moment for me.' |

Continued

**Table 2** Continued

| Identifier | Source | Quote |
|---|---|---|
| 15 | Participant 7, FG1 | 'I had a couple of different GPs that I spoke to at the beginning and then I spoke consistently to the same locum GP and she was very good. It was when I was having quite a difficult time trying to go back to work and I was struggling quite a lot psychologically and she was very supportive, she spent a lot of time with me and that consistency was good.' |
| 16 | Participant 13, FG2 | 'I've treated stroke patients who (have) dysphasia and they can't find the right words so they go around the houses to describe something so that you understand what they mean and it felt a bit like that in a way that you know what you want to say but you can't think what that word is because it doesn't come to the forefront of your mind. So you're trying to think of how you can describe it and I thought 'oh gosh, I've turned into one of my stroke patients' because I'm trying to find another suitable word but it's such a struggle though.' |

FG, focus group.

## Navigating the healthcare system

Participants had varying experiences of healthcare systems, with impaired memory and word-finding issues adding to the challenge of communication and self-advocacy (quote 13). Moreover, articulating the specifics

---

**Box 1   Definitions**

**Planning:** the mental process allowing individuals to choose necessary actions to reach a goal, ascertain the required order, assign tasks to cognitive resources and establish a plan of action.

**Decision-making:** the cognitive process resulting in the selection of a belief or a course of action from multiple possible alternative options.

**Flexibility:** the mental ability to adjust activity and content of the cognitive system, that is, enabling a switch between different task rules and corresponding behavioural responses, maintaining multiple concepts simultaneously and shifting internal attention between them.

**Complex attention:** a person's ability to maintain information in their mind for a short time and to manipulate that information, for example, to perform mental arithmetic calculations.

**Selective sustained attention:** the ability to focus on an activity or stimulus over a long period of time even if there are other distracting stimuli present.

**Divided attention:** the ability to attend to multiple different stimuli at the same time, thus responding to more than one demand from the surroundings, that is, enabling multitasking.

**Processing speed:** the time it takes a person to do a mental task, that is, the time at which a person can understand and react to the information they receive from sensory inputs and generate a reaction.

**Working memory:** a cognitive system with a limited capacity, capable of temporarily holding information to enable reasoning and guiding decision-making and behaviour.

**Procedural memory:** a type of implicit memory that aids the performance of particular types of tasks without conscious awareness of previous experiences, for example, stored motor programmes of particular well-rehearsed actions.

**Autobiographical memory:** a memory system formed from episodes recollected from an individual's life that combines episodic (personal experiences and specific objects, people and events experienced at particular time and place) and semantic (general knowledge and facts about the world) memory.

**Free recall:** a common memory task requiring individuals to recall any items from a previously memorised list either immediately or following a delay.

**Cued recall:** as above, individuals are required to recall items from a previously memorised list but may be given cues (often semantic) to encourage this.

---

of the 'brain fog' experience to healthcare professionals was a particular issue, and frustration, anger and hopelessness were commonly experienced when the impact of neurocognitive symptoms was 'downplayed', dismissed as being all 'in your head' or secondary to depression or anxiety, or deprioritised relative to other COVID-19 sequelae. Some participants felt that the fact they were middle-aged and female contributed to health professionals not taking them seriously.

Conversely, some participants described huge relief and validation at feeling believed and acknowledged (quote 14), particularly in the context of continuity, wise counselling and healthcare professionals bearing witness within therapeutic relationships (quote 15). Several participants had undergone brain imaging or neuropsychological testing, which were overwhelmingly normal and thus often enabled participants to focus on self-management, frequently supported by allied health professionals, including occupational therapists and physiotherapists. None reported having seen a psychologist or psychiatrist in any context.

## DISCUSSION
### Summary of key findings

This qualitative study of 50 UK participants suffering from neurocognitive symptoms following COVID-19 has revealed several important findings. Subjective impairments in executive function, attention, memory and language were common, often emerging from weeks to months after the acute illness and in most cases following a relapsing–remitting course that gradually improved over months. Prominent fatiguability and interaction between perceptually cognitive or physical symptoms combined with the impact on professional and personal activities, functional ability and identities to produce a destabilising, debilitating, frustrating, stigmatising and frightening situation. Variably successful approaches to mitigate the effect of brain fog included activity trade-offs and communication strategies, and the experience of illness was greatly compounded by the challenges in navigating the healthcare system when subjectively cognitively impaired.

## Comparison with theoretical literature

Some accounts of the condition fitted Frank's definition of the 'chaos narrative', where the illness experience is unresolved by restitution of the former healthy self, thus remains confusing and lacking in meaning.[44] The profound impact of symptoms on individuals' independence, self-efficacy and self-trust resonated with descriptions of spoiled identity and the disrupted sense of purpose and self that can accompany chronic illness,[51] while others aligned with theoretical accounts of shame and blame in other partly invisible conditions such as epilepsy.[52]

Participants' concerns also reflected the phenomenon of 'hidden disability', whereby individuals must undergo a contextual negotiation about when to 'pass' as able-bodied and when to self-identify as having a disability. In so doing, they must weigh up conflicting drivers of self-identity and preservation of self, impression management, stigma and legitimisation of or possible value judgements based on illness-related behaviour.[53 54] Moreover, the relapsing–remitting time course of brain fog symptoms also align with 'episodic disability', as described by those with HIV, to describe unpredictable periods of wellness and illness,[55] which adds an additional element of uncertainty.

Such requirements emphasise the extensive work people with long COVID-19 must undertake to manage their condition and navigate services, according to the theories of illness burden,[41] which, until recently, has been compounded by the lack of clear care pathways.[4] Positive experiences of care described dimensions of good professional practice: active listening and bearing witness,[48 56] wise counsel[47] and continuity of the therapeutic relationship[49] that alleviate patients' illness burden and help begin to construct a healing narrative.

Lack of understanding about the cause of neurocognitive symptoms was a frequent frustration for participants. Ongoing research has hypothesised neuronal damage occurs secondary to direct viral neurotoxicity[57] or associated neuroinflammation that generates a multisystem dysfunction resulting from a loss of central control and generalised peripheral inflammatory response.[12] Such suggestions are supported by pathological evidence of SARS-CoV-2 neurotropism[58] and neuroinflammation[59] combined with animal models of SARS-CoV-2 infection leading to neuroinflammation, intracellular Lewy body formation or neuronal loss.[29 60] It has been hypothesised that such processes impacting on vulnerable brain regions could correlate with neurocognitive symptoms in ongoing COVID-19 or post-COVID-19 syndrome: dysfunction of the brain stem, which is involved in regulation of both respiration and arousal—and thus potentially 'brain fog'—could account for some of the attentional deficits and disproportionate breathlessness seen in post-COVID-19 syndrome.[38 61] All of these theories need further research and correlation with the lived experiences reported in this study.

Finally, our findings illustrate that whatever the explanation for ongoing neurocognitive symptoms, the resultant impacts result from—and contribute to—a wider interplay of psychological, physical and social factors. The clear disruption to an individual's professional self, interpersonal relationships and overall sense of identity, combined with hidden and episodic disabilities, impairs sufferers' abilities to achieve Tarlov's anticipated state of 'health', described as 'the capacity, relative to potential and aspirations, for living fully in the social environment'.[62] Given that post-COVID-19 syndrome seems more prevalent among those of working age, in education[63] and particularly exposed 'key worker' groups,[63] the potential impact on society is significant. Therefore, while further work must deepen and exploit our mechanistic understanding, commissioners and providers of post-COVID-19 services, individual clinicians and employers must remain cognizant of the disruption to these broader components of health and consider how they may be mitigated to aid recovery.

## Strengths and limitations of the study

To the best of our knowledge, to date, this is the largest, most in-depth qualitative study of neurocognitive symptoms of post-COVID-19 syndrome. The research team included clinicians and social scientists. Our participants spanned a range of ages, ethnicities, social backgrounds and illness experiences—including the majority who were never hospitalised and a range of recovery states. Importantly, recovery state did not seem to affect individual perceptions or recollections of brain fog, which were described consistently. The majority of our participants were infected during the initial pandemic wave, thus email follow-up almost 12 months post-infection gives a meaningful insight into the condition's natural history. We oversampled men and people from non-white groups to partially correct an initially skewed sample. The use of multiple linked sociological theories allowed rich theorisation of the lived experience of the illness, supported by input from experts by experience.

The study does have limitations. The entirely UK-based sample included a high proportion of people recruited from a support group for those with neurocognitive symptoms of long COVID-19, thus likely to be more severely affected and potentially suffering from higher levels of distress.[64] Moreover, our sample did not extend to all demographic subgroups, so we may not have fully captured the perspectives of some minority ethnic groups, occupational classes or those less digitally connected. In the time since the first wave, knowledge and treatment of acute COVID-19 and post-COVID-19 syndrome have altered substantially with medical research, patient advocacy and (geographically variable) service development, which may influence the experience of long COVID-19 for people infected at later time points.

### Comparison with previous empirical studies

Our findings of persistent, debilitating neurocognitive symptoms in people living with long COVID-19 align with several retrospective cohort studies[18] and online patient surveys.[7 9 65 66] Our study adds further context to explore the functional and psychosocial impact of such symptoms and mitigating efforts by patients.

Comparisons have been made between post-COVID-19 syndrome and other post-infective syndromes of neurocognitive dysfunction. Infection with SARS-CoV-1,[67] Epstein-Barr virus, *Coxiella burnetii*, Ross River virus,[68] and *Borrelia burgdoferi*[69] can be associated with similar impairments to concentration and memory, typically correlated with persistent fatigue, although the causality of this association has been disputed. This study was not designed to compare the symptomatology of neurocognitive symptoms in people with long COVID-19 to other conditions. However, the challenge of unpicking the aetiology of brain fog is illustrated by the example of chronic fatigue syndrome/myalgic encephalomyelitis (CFS/ME), where persistent difficulties with executive function, short-term memory, attention and word-finding are incorporated in the diagnostic criteria of both the UK National Institutes for Clinical Excellence,[70] US Centers for Disease Control and Prevention[71] and International Consensus Group,[72] but where the cause(s) of these symptoms remain unclear.[73]

Examples such as HIV-associated neurocognitive dysfunction, which afflicts over 40% of people with chronic HIV infection,[74] impairing learning, memory, attention and executive function, suggest possible overlap across multiple chronic viral infections. A recent study in *Nature* illustrates how such higher order disruptions may be mediated on a molecular level through viral-associated perturbations in general cellular functions such as cortical excitatory synaptic signalling, choroid plexus disruption enabling peripheral T cell infiltration and promotion of pathological microglial and astrocyte subpopulations.[75] All of these mechanisms—and others—will require further elucidation.

Both the partially hidden nature of neurocognitive symptoms and the extensive work required to manage these and navigate services may contribute to the ongoing dispute about how common persistent symptoms are following COVID-19 infection. Data from the Office for National Statistics have demonstrated that self-reported long COVID-19 was greatest in people aged 35–69 years, women, people living in the most deprived areas, those in health and social care occupations, and those with another activity-limiting health condition or disability.[63] As for the acute infection, long-term sequelae of COVID-19 infection are strongly impacted by socioeconomic determinants such as poverty and structural inequalities, such as racism and discrimination,[76] which may affect health beliefs, health-seeking behaviours or the response of health services. While not directly reported by participants in this study, further work to explore the impact of such determinants on long COVID-19 epidemiology and interactions with health services will be crucial to mitigate the impact of associated disability.

### Conclusion: implications for services and further research

In dealing with COVID-19, it is crucial that health policy begins to shift from an acute disaster response to chronic crisis management. This study brought neuroscientists and qualitative researchers together to align the subjective illness experience with the perception of neurocognitive symptoms and proposed causal and contributory hypotheses. The profoundly disabling, persistent impacts of post-COVID-19 syndrome in a minority of people add weight to arguments that prevention of COVID-19 reduces not only mortality but also the long-term burden of disease on patients, the health service and the wider economy. Moreover, a better understanding of the pathophysiological mechanisms and further exploration of the best approaches to support cognitive, psychological and occupational restoration is crucial to aid those already affected.

The strong positive and negative emotional touchpoints[46] described by individuals when their accounts are, respectively, perceived as acknowledged or dismissed underscore the importance of the clinical relationship in which the patient is listened to, their experience believed and supported—particularly in primary care, which is likely to be the patient's first point of contact.[77 78] Furthermore, the varied nature of the severe impacts of neurocognitive symptoms identified in this study highlight the importance of ensuring that specialist services are accessible, easily navigable, comprehensive and interdisciplinary—for example, incorporating (where necessary) assessment and rehabilitation from clinical psychologists, cognitive neurologists and occupational therapists.[5] Our findings affirm those of a previous study to co-design quality indicators for post-COVID-19 syndrome services, which emphasised the importance of continuity, clinical responsibility, multidisciplinary input, patient involvement and use of evidence-based guidelines.[4]

**Acknowledgements** We thank the 50 participants for their interest and contributions, and 2 experts by experience for helpful comments on a draft of the paper: Sharon Taylor, child psychiatrist and honorary senior lecturer at the Central and North-West London NHS Foundation Trust and Imperial College School of Medicine, and Clare Rayner, Independent Occupational Physician, Manchester. Alex Rushforth and Sietse Wieringa undertook interviews for the original study of long COVID.

**Contributors** el and TG conceptualised and designed the study. el, LH, CC and KP conducted focus groups. el and CC led data analysis, with input from LH, TG and KP and produced a first draft of the results section. el and CC wrote the first draft of the paper, which was refined by all authors. LH provided research assistant support and conducted some interviews. Sharon Taylor and Clare Rayner provided expertise by experience and knowledge of patient-led research. CC presented findings to long COVID-19 patient participants with assistance from el and TG. All authors contributed to refinement of the paper and provided additional references. el is the corresponding author and guarantor and affirms that the manuscript is an honest, accurate and transparent account of the study being reported, that no important aspects of the study have been omitted and that any discrepancies from the study as planned (and, if relevant, registered) have been explained.

**Funding** This research is funded from the following sources: National Institute for Health Research (BRC-1215-20008), ESRC (ES/V010069/1) and Wellcome Trust (WT104830MA). Funders had no role in the planning and execution of the study or writing up of the paper. KP is supported by the National Institute for Health Research Biomedical Research Centre based at Oxford University Hospitals NHS Foundation Trust and the University of Oxford.

**Competing interests** el and TG provided evidence on long COVID-19 for House of Lords Select Committee. TG was on the oversight group for the National Institute for Health and Clinical Excellence guideline on managing the long-term effects of COVID-19, and at the time of writing was on the UK's National Long COVID Taskforce.KP and CC have no competing interests to declare.

**Patient and public involvement** Patients and/or the public were involved in the design, or conduct, or reporting, or dissemination plans of this research. Refer to the Methods section for further details.

**Patient consent for publication** Not applicable.

**Ethics approval** This study involves human participants and was approved by ethical approval was granted from the East Midlands–Leicester Central Research Ethics Committee (IRAS Project ID: 283196; REC reference number: 20/EM0128) on 4 May 2020 and subsequent amendments. Participants gave informed consent to participate in the study before taking part.

**Provenance and peer review** Not commissioned; externally peer reviewed.

**Data availability statement** Data are available upon reasonable request. Deidentified participant focus group data may be available from the corresponding author, using the correspondence contact details. This is subject to the correct ethical approvals and data sharing approvals and data governance structures being in place.

**ORCID iDs**
Emma Ladds http://orcid.org/0000-0001-9864-7408
Trisha Greenhalgh http://orcid.org/0000-0003-2369-8088

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
