## [Reviewer comments · BMJ Open]

ARTICLE DETAILS

TITLE (PROVISIONAL)	"I can't cope with multiple inputs": Qualitative study of the lived experience of 'brain fog' after Covid-19
AUTHORS	Callan, Caitriona; ladds, emma; Husain, Laiba; Pattinson, Kyle; Greenhalgh, Trisha

VERSION 1 – REVIEW

REVIEWER	Vogt, Henrik Norges teknisk-naturvitenskapelige universitet I am an MD and postdoctoral researcher in medical scientific theory at the Centre for medical ethics in Oslo. I'm also the leader of Recovery Norway, an organization consisting of people who have recovered from disorder such as ME/CFS and are similar to post-covid syndrome through strategies involving changes in e.g. behaviour and cognition. As such I have an intellectual conflict of interest when it comes to how such disorders may be mitigated.
REVIEW RETURNED	25-Aug-2021

GENERAL COMMENTS	IN MY COMMENTS I USE THE PAGE NUMBERS ON THE UPPER RIGHT CORNER OF THE PAGES IN THE PROOF Overall this is an important paper, addressing important question with an appropriate methodology. I think it should be accepted but with major revisions. My most important criticism is this: The authors want to align the subjective illness experience and objective disease models. This is an interesting and important project! However, there is a complete lack of reference to the science of how symptoms persist. There is already a large field that has tried to make this connect for a long time and provides a rich and indispensable theoretical background. I don't think it is acceptable to produce a paper with no reference to an entire field addressing the problem. The authors should know and use this literature. I think the authors should be required to refer to these and utilize them. Otherwise, this article does not have a foothold in existing knowledge. This is what I mean by major revision here. I suggest as examples of this literature: https://www.sciencedirect.com/science/article/abs/pii/S0272735820300179 and https://core.ac.uk/reader/80805780?utm_source=linkout and https://pubmed.ncbi.nlm.nih.gov/28856337/ Also: The article wholly lacks a clear explication that "brain fog" is not a scientific term and that it is known in a range of other conditions. An impression is created that this is something unique to post covid syndrome. For example, it is well-known among sufferers of anxiety or stress, PTSD, fibromyalgia, CFS/ME ++. See below. Again, without reference to this literature this article builds on thin air where there is actually knowledge to stand on.
---

ABSTRACT

- p 4 LINE 10. Brain fog is here defined as “the wide variety of neurocognitive symptoms that can follow covid-19”. This is a mistake that runs through the whole paper. It is important from the outset – and throughout the text - to be precise and make it clear that “brain fog” is not specific to Covid-19, but is a well-known symptom of subjectively experienced cognitive impairment. The definition is not adequate.
- p 4, 10: Terms like “neurocognitive symptoms” is used in fields such as neuropsychology, but are misnomers. There can be no symptom or cognition that is not also “neuro” as that would require a disembodied soul. Use conceptually sound language. “Symptom” will do. Symptom is the first-person experience of the body.
- P. 4, 33: “for people with persistent neurological problems”. It is true that every symptom is also “neurological” in a wide sense. But brain fog is not “neurological” in the sense that “neurological” most often is used. “Neurological” then means the things neurologists deal with, which is “organic” as opposed to “functional”, meaning things that are objectively damaged and can be expected to stay damaged through time. So, the participants in the online did not likely have “neurological” problems as commonly understood. I suggest the authors should stick with descriptive, precise terminology: It is about people with persistent symptoms of e.g fatigue, cognitive problems after an infection.
- P. 4, line 33: “Long covid” is a term but was invented by patient groups, but that is nonetheless inappropriate term for science and academic. “Long covid” is a misnomer suggesting that there is a long-lasting covid-19 disease. Covid-19 is a viral disease. There is little or no evidence to suggest that the symptoms are due to long-lasting viral disease, and one should not incorporate such a preconception into the terminology. We do not talk about “long Epstein barr” when we talk about symptoms after that or “long influenza”. We talk about post-viral syndromes. Talk about “chronic Lyme” is an example how such terminology is deeply confusing and inappropriate (see e.g. Feder et al., NEJM, 2007). That “everybody” now says long covid is no excuse or good reason to use it. Post-covid syndrome, also used by NICE, SIGN and RCGP is more adequate. Please change the terminology or write something like “post covid syndrome”, also known as “long covid”. The researchers should abstain from this term and instead use the more causally neutral “post covid syndrome”, what we are phenomenologically dealing with here are symptoms after a covid infection. We do not know that the causal factors that prolong the duration of symptoms are the same as the initial causes, as is evident in e.g. pain research and other research on prolonged symptoms.
- P. 4, line 46: The analysis was “checked” by people with experience of brain fog. Were these people still experiencing this? Or had they recovered. People who have recovered will often have very different experiences than the chronically ill and different things to “check”.

BACKGROUND

- P 8, 6. Opening line. It is incorrect that it is “well established” that covid-19 can *cause* symptoms beyond the acute phase. It can INITIATE such symptoms, like many other acute conditions, but as is evident in research on this the causes that lead to chronification and “long” trajectories may be different. This is an empirical question, and the authors are incorrect to state that this is well-known. One could correctly write that covid-19 can *initiate* such symptoms although the causes of the persistence are empirically investigated.
- P 8, 6-27: Again, terms like “long covid” and “chronic covid” are conceptually inadequate and misleading and the authors should use the more descriptive “post covid syndrome”.
- P. 8, 46-54: There is very little to suggest that persistent viremia is the cause in the vast majority of people. This is reminiscent of chronic lyme

	activism where there is an ongoing pressure to understand symptoms as a manifestation of a “hidden” infection. ... - P. 8, 46-54: “Psychological factors”. In a world where we do no longer accept a mind-body dischotomy, what do the authors mean be “psychological”? This is not a trivial question as these concepts are used throughout. Please add to the list of definitions or use more descriptive language about what is actually meant. - P. 9, 3-19: NB! In the background and throughout the article, there is a complete lack of reference to the science of how symptoms persist. This is not adequate. There is a large scientific body on this. I suggest as examples: https://www.sciencedirect.com/science/article/abs/pii/S0272735820300179 and https://core.ac.uk/reader/80805780?utm_source=linkout and https://pubmed.ncbi.nlm.nih.gov/28856337/ - P. 9: It should be noted that these symptoms of post-covid are very rare if existent at all in smaller children, which probably says something about etiology too. - P. 9, 20-42: It is not ok exclusively to refer to “pathophysiologies” in a reductive mechanistic sense. It is not representative of science and it creates an unnecessarily scary picture of the problems. As evident in the literature linked to above, and also now in empirical studies on post-covid symptoms specifically, cognitive and behavioral aspects are important. See some example references which should be considered for inclusion: https://www.medrxiv.org/content/10.1101/2021.06.25.21259256v1 AND https://journals.sagepub.com/doi/full/10.1177/14034948211018385 AND https://journals.plos.org/plosone/article?id=10.1371/journal.pone.0240784 see also https://academic.oup.com/oim/article/2/1/iqab004/6131647 - P. 9. An important reference is lacking: https://www.thelancet.com/article/S1473-3099(21)00211-5/fulltext - P. 9, line 49. A distinction should be made between the impairment people experience and their impairment as assessed by e.g. cognitive tests. That people experience that they function poorly may be mediated by e.g anxiety and their meaning-making/interpretation and often does not equate with testing of cognitive abilities. - P. 10, line 8. The use of the term “mechanistic underlying...” suggests a causally reductive preconception that the best explanations will be found on the molecular level “underlying” everything else. Stick to something less preconceived? What is meant is cause more generally. - P 10: Research question: Again, terms like “neurocognitive symptom” or “psychocognitive” are nonsensical as there are no symptoms that are not “neuro” og no “cognitive” that is not also “psychological. Consider using simple, adequate, descriptive terms: cognitive process. Just symptom. - P. 10: NB! The background wholly lacks a clear explication that “brain fog” is not a scientific term and that it is known in a range of other condtions. What is it really? What are the scientific terms with references? “Brain fog” is nothing very mysterious and it is strange and inadequate to act as is science has not studied this before. The authors risk mystifying – making us less knowledgeable than we really are – about something that we know something about. For example, it is well-known among sufferers of anxiety. See e.g. https://www.anxietycentre.com/anxiety-disorders/symptoms/brain-fog/ . It is very problematic that researchers of post-covid fatigue write as if this is something new or unique to post-covid. It is also known as e.g. cognitive impairment. Try for example searching for “cognitive impairment” and stress. It will easily reveal many references both from science and people’s experiences and online fora. It is seen in e.g. fibromyalgia, CFS/ME and PTSD. See e.g.... https://www.psychiatryadvisor.com/home/topics/neurocognitive-disorders/alzheimers-disease-and-dementia/stress-increases-the-risk-of-mild-cognitive-impairment/ OR https://www.verywellhealth.com/brain-fibro-fog-causes-symptoms-possible-treatment-716014 OR . It is totally
--	--

unnecessary and misleading that the authors write as if we know little about this phenomenon from before. The background is in clear need of an update on the science of persistent symptoms and “brain fog” or e.g. cognitive impairment more generally.

METHODS

- P 12, line 10. Although frequently used, the terms “physical symptom” is nonsense (in a philosophical sense of being conceptually inadequate). There are no symptoms that are not subjective or “psychological” per definition. I take it that the authors do not really mean that cognitive symptoms do not refer to something physical (the brain). If the authors are going to use such conceptual language, they should define what they mean by a “physical” symptom as opposed to a “neurocognitive” symptom. Please add in the definition list.

THEORETICAL FRAMEWORK

- P 12, line 55. A definition of a neuroscience perspective is NOT to think that the cause needs to be SARS-COV2 disrupting the brain and the brain stem. Neuroscience can be said to be a reductive (and productive) endeavor trying to reduce e.g. symptoms to brain function, but such a perspective does NOT necessarily involve thinking that SARS-COV2 disrupts or is the cause of the problems in itself. This is a pretty serious theoretical mistake and statement. The reference to allostasis is a good one, neuroscience is not just about the brain. However, there is no theoretical divide between allostasis and homeostasis. Allostasis is about keeping homeostasis through dynamic change. It is faulty to state that systems of homeostasis or allostasis AND physiology INTERACT with brain systems of mood etc. Systems of mood, attention etc of course is PART OF the body’s allostatic systems. The allostatic theoretical perspective which seeks to overcome the divide between “psychological” and the body is largely missing in the text further on (McEwen).

- P. 13. NB! I think the theoretical perspective is sorely lacking a reference to the science of persisting systems. I think the authors should be required to refer to these and refer to them. There is a large body of science trying to explain just what the authors are dealing – the bridge between the perspective of neuroscience and the perceived symptoms - with here and they don’t refer to it. I can’t see that as adequate. See again <https://www.sciencedirect.com/science/article/abs/pii/S0272735820300179> or https://core.ac.uk/reader/80805780?utm_source=linkout

RESULTS

- P. 14-15 lines 50 and onward: It may seem here to the reader as if the patients have been subjected to a neuropsychological examination. They have not. The authors do not know that their experiences would match “objectively” assessed cognitive function. This should be made clear. Their experiences would correspond to these domains and MAY correspond to such deficits but their actual lack of function in these domains have NOT been assessed.

- P 15, line 54: “Physical fatigue”. This is again an example of a concept that is nonsensical in a world without a mind-body dichotomy. What is “physical fatigue” exactly? What is the difference from “mental fatigue”? There is only one fatigue. If the authors think there is a difference and wish to use these messy terms they should define what they mean by them. The authors should explain what they mean by this or just use descriptive terms. What symptoms are we talking about?

- P16. The study lacks a reference to something very important: What does the “fog” mean in the particular individual’s life context. For example, someone working with something that demands a lot of cognitive attention or multitasking may feel differently than those who do not.

- P 17, lines 18-30. The sense that one is “validated” if the brain fog is thought to be due to e.g. brain damage and that it is somehow non-valid that it is due to anxiety or depression or “in your head” is an important finding. The authors could elaborate that, do the subjects say more? However, the authors should watch their language and make sure that THEY don’t suggest and validate the idea that brain fog due to anxiety or depression is somehow second range or non-valid and thus contribute to the stigma against disorders like these (which likely underlies the respondents’ feelings of anger). The authors should not suggest that healthcare workers talking to patients about how anxiety or fatigue can cause brain fog is not validating them. This is disrespectful towards the sufferers of these conditions who suffer something every bit as real as anything else.

- P 17: Hypothesising mechanisms. The use of the term “mechanisms” is not neutral theoretically. By mechanism we often understand something on the molecular level or low biological level. What is hypothesized here is not mechanism, which the human experience cannot access through our senses, but causes more broadly. Using analogies of stroke for example is not about hypothesizing cause in a scientific sense, it is about the ideas these people have of their illness, their meaning-making. The theories they put forward are not taken from their own experiences, they are taken from something they have read about.

-
DISCUSSION

- P 18: Discussion: The authors here claim to have revealed that the symptoms “included deficits”. They have not shown this. They have documented symptoms that MAY correspond to such deficits, but they have not performed any tests to check if these perceptions correspond to a deficit in these neurocognitive domains. This should be made perfectly clear.

- P. 18: line. Statements such as “interaction” between physical and cognitive symptoms are again philosophically nonsensical as there are no symptoms that are non-physical or non-cognitive. What do the authors actually mean? Please state this in clear language.

- P 18, line 47: The authors take it as granted that the patients lacked neurocognitive skills. This is not something the authors know. This is an illness *perception* or *belief*. It can be, but is not *necessarily* so, that they perform very bad on tests.

CONCLUSION:

- P 20, line 6-23. In the middle of what is an account of the experiences of people with the disorders the authors venture into an account of what ongoing research is supposed to show about pathophysiological cause. This small discussion is heavily skewed towards a reductive (and scary) view which completely omits both prior knowledge about such symptoms (which the authors should familiarize themselves with) and specific references showing links to e.g anxiety, see above. More importantly, this passage is out of place in a discussion about patient experience. They are frustrated with having no reductive pathophysiological explanation which in biomedicine denotes something “real”. The authors don’t have to help them with such a reductive explanation. It is not their project or research question in this article. This passage should be deleted. More should be written about this experience, which is important. Why is it important for them to have a specific explanation?

- P 20: Results/Discussion: General: A strength is talking to people who have recovered. I would want more discussion about how the narratives of those who recover differ from those who do not. This is very important variation in the dataset which should be explored.

- P 21 Selection: The important problem with selection of people who frequent online fora as representative is noted discussion on limitations.

	The authors should refer to research showing that these are the patients who experience the most distress. https://pubmed.ncbi.nlm.nih.gov/32758507/  - P 22: NB! Again, the discussions lacks connection to the science of persistent symptoms - P 22, line 38. The authors state that infection with a list of microbes RESULT in the impairments. This has not been shown. The authors need to distinguish between initiating causes and causes that cause chronification. The causes of persistent infections are an empirical question. Borrelia burgdorferi does not RESULT in all these symptoms. On the contrary, the infectious disease community have long struggled to communicate that the symptoms experienced after Lyme disease are probably caused by other factors (see e.g. Feder et al., 2007, NEJM). - P 23: line 28: Again, the depiction of the disabilities as “neurological” is misleading as it suggests organic damage. - P 24: Conclusion. NB! Again, here the authors state something important about their objective: To align the subjective illness experience and objective disease models. This is connected to my most important criticism here: The lack of reference to and theoretical use of literature that already makes this connection and has studied it for a long time. See references above. Relatedly, it is a huge problem that the authors write as if brain fog is something specific or new to post-covid syndrome. It is not. - P 24: I think the authors should refrain from the political argument above preventing covid. It is not connected to their research question. It could also be noted that it is ethically tenous for researchers to contribute to anxiety and symptom focus about a condition, as this will likely increase the very symptoms and distress they are investigating (see Barsky, JAMA, 2017: The iatrogenic potential of physician’s words). - P 24: “Neurotropic” virus. It is unjustified to present this virus in this way. It has no special affinity for nerve cells. It is a respiratory virus with systemic effects (like many other viruses). Stick with “virus”. - P 25, line 35-37: It is not supported that the patients have actually not been “believed” or “supported”. This is about their experiences and their agency in portraying things. I think, again, the authors should avoid supporting the notion that connecting “brain fog” to e.g. anxiety it not about believing people. It is derogatory towards these forms of illness.
--	---

REVIEWER	Humphreys, Helen
REVIEW RETURNED	Sheffield Hallam University, Advanced Wellbeing Research Centre 11-Oct-2021

GENERAL COMMENTS	This paper addresses an important and interesting phenomenon using an appropriate methodology and a commendable sample size. Whilst the findings are interesting, I feel that some sections could be developed further to ensure that the paper adds to the wider understanding of brain fog. I hope my comments are helpful. Abstract The conclusions stated in the abstract are quite broad and could easily be mistaken for general recommendations about managing long Covid. Can these be more specifically linked to what the paper has highlighted about brain fog in particular? Background Could the authors add a brief, critical summary of previously published qualitative research on long Covid to strengthen the rationale for this in-depth exploration of brain fog?
---

	Methods Please clarify the consent process - were participants provided with written information about the study and the way their data would be used? How was consent obtained from each individual participant? How many researchers were involved in the data collection and analysis and how were the tasks divided? Under “data management and analysis” the term “we” is used but it isn’t clear to what extent investigator triangulation or similar took place. It’s great to see the theoretical lens for the research outlined in detail but would be useful to have a brief line to describe how this informed the analysis. Patient Involvement Statement It would be useful to have an indication about the types of modifications that were made to the draft paper based on participants’ feedback. Results Can a list of the final themes be added at the start of the results section? If possible, it would be easier to follow if the themes names were consistent across the abstract and the results section. To be honest, I found it difficult to follow the results with the participant quotes provided in a separate table. I acknowledge this may be due to word count limitations and also because some quotes are referred to more than once but if it’s possible to include them in the main text it would bring the themes to life much more (perhaps aim to provide different quotes rather than repeating some to make use of the “1000 pages of transcripts and notes” that were collected). Whilst the themes were relatively self-explanatory, I often found myself wanting to learn a little more. For example, Neurocognitive symptoms - was it common for participants to experience symptoms across different domains of cognitive function? A key point highlighted in other long Covid ‘lived experience’ research has been the interplay between physical and cognitive symptoms and the associated impact on managing fatigue and activity. Did the authors learn anything further about how participants felt that these elements of their condition affected one another? I found 9 themes a lot to read. I don’t wish to dictate what themes the authors choose to present, but I do wonder if a little more development/analysis might allow some of the themes to be combined into something more comprehensive - e.g. Psychosocial impact of neurocognitive symptoms / Guilt, shame and stigma; Hypothesising mechanisms / Self-management Navigating the healthcare system - the challenges described here echo previous lived experience research and these have been reasonably well documented to date. The particular challenges of communicating and self-advocating whilst experiencing brain fog are perhaps not emphasised enough. Were there any particular neurocognitive symptoms that were particularly likely to be dismissed by healthcare professionals? What sort of investigations had healthcare professionals been willing to offer or refer to? What
--	---

	was the role of long Covid clinics in this experience (or were these not yet established)? Discussion The opening paragraphs of the discussion section paint a compelling picture of how challenging the experience of brain fog can be which I am not sure has come through in the results section. The discussion could be strengthened with more specific discussion about the key points raised by participants. For example, the term “brain fog” was not considered to accurately describe some peoples’ experience. Is it appropriate to keep using this or might another term be more useful? What does this paper tell us about how to strike a balance between medical investigation of neurocognitive symptoms that helps to understand the causes and mechanisms against the application of self-help strategies? Is brain fog an inevitable consequence of long Covid that people with the condition will have to simply endure or are there specific things that either they, or people supporting them can do to make it more bearable and/or alleviate the challenges somewhat? How are neuropsychiatric symptoms currently being addressed (and by whom) within long Covid rehab pathways and what changes/recommendations to those pathways does this study suggest? The impact of these symptoms on peoples’ ability to return to work or maintain their regular employment schedules is not insignificant. Is there anything to be said about the types of roles/jobs/employees that are particularly impacted by this? Besides reducing hours, what reasonable adjustments could employers be making? Limitations The absence of people who are digitally excluded appears to be a recurring limitation in long Covid research - can the authors make any comment or recommendation about how future research might overcome this? It might also be prudent to acknowledge this in the strengths and limitations section at the start of the paper. The full paper requires a thorough proof-read and check for spelling and grammar.
--	---

VERSION 1 – AUTHOR RESPONSE

Reviewer: 1

Dr. Henrik Vogt, Norges teknisk-naturvitenskapelige universitet

Comments to the Author:

*** Please find supporting documents for this review attached to this email ***

We ask the editors to note this reviewer's self-declared "intellectual conflict of interest". He works a lot with people who have recovered from (and some who have failed to recover from) CFS and ME. He himself points out that his approach to long Covid is coloured by this perspective. We strongly agree. In many places in his review, he stridently points out that there are parallels between persistent Covid and CFS/ME. We're not sure he's correct here, and whilst he cites a lot of literature, we're not sure all of it is relevant. We do however greatly respect his insight that his own views may have been distorted by where he's coming from clinically and in his own research.

Dear authors.

IN MY COMMENTS I USE THE PAGE NUMBERS ON THE UPPER RIGHT CORNER OF THE PAGES IN THE PROOF

Overall this is an important paper, addressing important question with an appropriate methodology. I think it should be accepted but with major revisions. My most important criticism is this: The authors want to align the subjective illness experience and objective disease models. This is an interesting and important project! However, there is a complete lack of reference to the science of how symptoms persist. There is already a large field that has tried to make this connect for a long time and provides a rich and indispensable theoretical background. I don't think it is acceptable to produce a paper with no reference to an entire field addressing the problem. The authors should know and use this literature. I think the authors should be required to refer to these and utilize them. Otherwise, this article does not have a foothold in existing knowledge. This is what I mean by major revision here. I suggest as examples of this literature:

<https://www.sciencedirect.com/science/article/abs/pii/S0272735820300179> and

https://core.ac.uk/reader/80805780?utm_source=linkout and

<https://pubmed.ncbi.nlm.nih.gov/28856337/>

Also: The article wholly lacks a clear explication that "brain fog" is not a scientific term and that it is known in a range of other conditions. An impression is created that this is something unique to post covid syndrome. For example, it is well-known among sufferers of anxiety or stress, PTSD, fibromyalgia, CFS/ME ++. See below. Again, without reference to this literature this article builds on thin air where there is actually knowledge to stand on.

We accept this criticism up to a point, and we have now included a brief literature review in the background section which mentions the papers cited above (pages 7-8). However, we respectfully point out that

a) our study is conducted in the tradition of patient-centred research, and deliberately use the term 'brain fog' because that's what patients call it. To say the term is 'unscientific' is a somewhat pejorative response! We'd say the term is subjective and meaningful to patients, though we agree (indeed, we describe on page 8) that it doesn't have a precise scientific definition and we explain the distinctions from more rigorously defined terms. We suggest that BMJ Open might like to ask the advice of a patient reviewer on whether these 'patient-made' terms should be expunged from academic papers (Perego et al have written very well on this topic (1)).

b) BMJ Open has a fairly tight word count and a comprehensive review of the theoretical literature would not be possible in addition to detailed empirical findings,

c) to our knowledge, the claim that the brain fog of long Covid is the same as the brain fog of the other conditions listed above is a hypothesis not an established fact. To compare the symptom patterns across conditions would require a different study design. What we have done is describe brain fog in long Covid patients, based on their own accounts of lived experience. We have, however, amended the document to point out that the brain fog of long Covid may have parallels to

that of other conditions, and suggested that this would be a fruitful area for further research (pages 11 and 18).

ABSTRACT

- p 4 LINE 10. Brain fog is here defined as “the wide variety of neurocognitive symptoms that can follow covid-19”. This is a mistake that runs through the whole paper. It is important from the outset – and throughout the text - to be precise and make it clear that “brain fog” is not specific to Covid-19, but is a well-known symptom of subjectively experienced cognitive impairment. The definition is not adequate.

See above – we believe this is the reviewer’s opinion but that there is not yet scientific proof either way. As noted above, we’ve acknowledged the possibility that there may be fundamental similarities but we believe it would be unscientific to treat this assertion as a known fact.

- p 4, 10: Terms like “neurocognitive symptoms” is used in fields such as neuropsychology, but are misnomers. There can be no symptom or cognition that is not also “neuro” as that would require a disembodied soul. Use conceptually sound language. “Symptom” will do. Symptom is the first-person experience of the body.

The term ‘neurocognitive’ is widely used by medical authors (it comes up 36581 times on PubMed for example). The reviewer is right to push us to be clear about what we mean by it. We use the US National Cancer Institute definition, namely “Having to do with the ability to think and reason. This includes the ability to concentrate, remember things, process information, learn, speak, and understand.” This is exactly what we mean by the term and we have included this definition on page 8 to better differentiate what we mean by its use. We do not state or imply a disembodied soul. “Symptom” is of course a much broader term and will not do.

- P. 4, 33: “for people with persistent neurological problems”. It is true that every symptom is also “neurological” in a wide sense. But brain fog is not “neurological” in the sense that “neurological” most often is used. “Neurological” then means the things neurologists deal with, which is “organic” as opposed to “functional”, meaning things that are objectively damaged and can be expected to stay damaged through time. So, the participants in the online did not likely have “neurological” problems as commonly understood. I suggest the authors should stick with descriptive, precise terminology: It is about people with persistent symptoms of e.g fatigue, cognitive problems after an infection.

We accept this criticism but think the best way to deal with it is use the term neurocognitive here, as defined above and we have corrected the paper throughout eg: in the abstract pages 3-4.

- P. 4, line 33: “Long covid” is a term but was invented by patient groups, but that is nonetheless inappropriate term for science and academic. “Long covid” is a misnomer suggesting that there is a long-lasting covid-19 disease. Covid-19 is a viral disease. There is little or no evidence to suggest that the symptoms are due to long-lasting viral disease, and one should not incorporate such a preconception into the terminology. We do not talk about “long Epstein barr” when we talk about symptoms after that or “long influenza”. We talk about post-viral syndromes. Talk about “chronic Lyme” is an example how such terminology is deeply confusing and inappropriate (see e.g. Feder et al., NEJM, 2007). That “everybody” now says long covid is no excuse or good reason to use it. Post-covid syndrome, also used by NICE, SIGN and RCGP is more adequate. Please change the

terminology or write something like “post covid syndrome”, also known as “long covid”. The researchers should abstain from this term and instead use the more causally neutral “post covid syndrome”, what we are phenomenologically dealing with here are symptoms after a covid infection. We do not know that the causal factors that prolong the duration of symptoms are the same as the initial causes, as is evident in e.g. pain research and other research on prolonged symptoms.

See above. We use the term long Covid very deliberately, because this is a qualitative paper about the patient's lived experience. TG was on the NICE guidelines oversight group that rejected the term “long Covid” in favour of “post-acute” and “chronic” Covid. The reason the panel chose to do that (and there was much debate) was that they wanted to base the terminology on ‘objective’ criteria. This is all good and rational. But this does not mean the subjective illness ceases to exist. We also need a name for what patients call their experience. Otherwise we are effectively saying to patients, “your illness is what I measure”. It's a bit like the joke about the patient who goes into the doctor's clinic with a sore tongue and comes out with glossitis. But glossitis is NOT a sore tongue. Some people will have a sore tongue with no objective evidence of glossitis and vice versa. So because in this study we're studying the patient experience, we're using the correct (ie. patient-made) term for that.

We have carefully reviewed the paper throughout to ensure we are only using ‘long Covid’ to refer to individuals’ subjective experience and ‘post-Covid-19 syndrome’ where we refer to the broader societal or scientific view of the illness.

- P. 4, line 46: The analysis was “checked” by people with experience of brain fog. Were these people still experiencing this? Or had they recovered. People who have recovered will often have very different experiences than the chronically ill and different things to “check”.

We've made this clear on page 12 in the patient involvement statement.

BACKGROUND

- P 8, 6. Opening line. It is incorrect that it is “well established” that covid-19 can *cause* symptoms beyond the acute phase. It can INITIATE such symptoms, like many other acute conditions, but as is evident in research on this the causes that lead to chronification and “long” trajectories may be different. This is an empirical question, and the authors are incorrect to state that this is well-known. One could correctly write that covid-19 can *initiate* such symptoms although the causes of the persistence are empirically investigated.

We accept this criticism and have amended the wording on page 7 to “well-established that symptoms can occur beyond the acute Covid-19”.

- P 8, 6-27: Again, terms like “long covid” and “chronic covid” are conceptually inadequate and misleading and the authors should use the more descriptive “post covid syndrome”.

Throughout the paper, we've used the term ‘long Covid’ to refer to the patient experience and ‘post covid-19 syndrome’ to refer to a medically diagnosed condition. In most places this means we keep the term ‘long Covid’ however, we have carefully scrutinised the document to ensure that these terms are used in the appropriate contexts.

- P. 8, 46-54: There is very little to suggest that persistent viremia is the cause in the vast majority of people. This is reminiscent of chronic lyme activism where there is an ongoing pressure to understand symptoms as a manifestation of a “hidden” infection. ...

We respectfully point out that the wording of the paragraph was as follows: “Whilst the underlying pathophysiology remains unclear, persistent viraemia [12], relapse or reinfection [13] inflammatory and immune reactions [14, 15], deconditioning [16] and psychological factors [17, 18] have all been proposed as contributors.”

We do not want to get into a long discussion about the evidence for or against persistent viraemia – we’re just stating the fact that persistent viraemia is ONE possible mechanism that has been discussed in peer-reviewed published papers. We don’t single it out, and we suspect a different reviewer might take issue with (say) deconditioning or psychological factors as a cause.

- P. 8, 46-54: “Psychological factors”. In a world where we do no longer accept a mind-body dichotomy, what do the authors mean by “psychological”? This is not a trivial question as these concepts are used throughout. Please add to the list of definitions or use more descriptive language about what is actually meant.

We agree, but this is what the paper called it.

- P. 9, 3-19: NB! In the background and throughout the article, there is a complete lack of reference to the science of how symptoms persist. This is not adequate. There is a large scientific body on this. I suggest as examples:
<https://www.sciencedirect.com/science/article/abs/pii/S0272735820300179> and
https://core.ac.uk/reader/80805780?utm_source=linkout and
<https://pubmed.ncbi.nlm.nih.gov/28856337/>

Thank you for these helpful references; we have added an additional section on page 11 in the discussion on theoretical frameworks to try and capture more of this concept. However, please note that the references provided are in a different disciplinary space to the literature on lived experience. They are not ‘better than’ patient accounts, they are complementary to those accounts and we have used alternative references that seem to correspond more with the neuroscience perspective being discussed here.

- P. 9: It should be noted that these symptoms of post-covid are very rare if existent at all in smaller children, which probably says something about etiology too.

We don’t talk about children in the paper at all. Our sample was >18s.

- P. 9, 20-42: It is not ok exclusively to refer to “pathophysiologies” in a reductive mechanistic sense. It is not representative of science and it creates an unnecessarily scary picture of the problems. As evident in the literature linked to above, and also now in empirical studies on post-covid symptoms specifically, cognitive and behavioral aspects are important. See some example references which should be considered for inclusion:
<https://www.medrxiv.org/content/10.1101/2021.06.25.21259256v1> AND
<https://journals.sagepub.com/doi/full/10.1177/14034948211018385> AND
<https://journals.plos.org/plosone/article?id=10.1371/journal.pone.0240784>
see also <https://academic.oup.com/oim/article/2/1/iqab004/6131647>

It seems like the reviewer does not want us to use the term “pathophysiological”. Whilst we respectfully suggest that the above paragraph is a bit of an over-reaction to the sentence “Lack of mechanistic understanding of the pathophysiological cause was a frequent frustration for participants”, we are happy to remove the word “pathophysiological” from it on page 9.

- P. 9. An important reference is lacking:
[https://www.thelancet.com/article/S1473-3099\(21\)00211-5/fulltext](https://www.thelancet.com/article/S1473-3099(21)00211-5/fulltext)

Thanks. Added as reference 31.

- P. 9, line 49. A distinction should be made between the impairment people experience and their impairment as assessed by e.g. cognitive tests. That people experience that they function poorly may be mediated by e.g. anxiety and their meaning-making/interpretation and often does not equate with testing of cognitive abilities.

We agree these impairments should be distinguished and have done so in the revised manuscript in the first paragraph on page 8 and the theoretical framework paragraph on page 11.

- P. 10, line 8. The use of the term “mechanistic underlying...” suggests a causally reductive preconception that the best explanations will be found on the molecular level “underlying” everything else. Stick to something less preconceived? What is meant is cause more generally.

We have amended the terminology on page 9 from ‘further knowledge of both the mechanistic aetiologies underlying such symptoms’ to ‘further knowledge of the underlying causal and contributory factors’.

- P 10: Research question: Again, terms like “neurocognitive symptom” or “psychocognitive” are nonsensical as there are no symptoms that are not “neuro” or no “cognitive” that is not also “psychological. Consider using simple, adequate, descriptive terms: cognitive process. Just symptom.

We’ve dealt with this above by defining what we mean by the term neurocognitive on page 8 and ensuring consistent and accurate use of this term throughout the manuscript.

- P. 10: NB! The background wholly lacks a clear explication that “brain fog” is not a scientific term and that it is known in a range of other conditions. What is it really? What are the scientific terms with references? “Brain fog” is nothing very mysterious and it is strange and inadequate to act as if science has not studied this before. The authors risk mystifying – making us less knowledgeable than we really are – about something that we know something about. For example, it is well-known among sufferers of anxiety. See e.g. <https://www.anxietycentre.com/anxiety-disorders/symptoms/brain-fog/> . It is very problematic that researchers of post-covid fatigue write as if this is something new or unique to post-covid. It is also known as e.g. cognitive impairment. Try for example searching for “cognitive impairment” and stress. It will easily reveal many references both from science and people’s experiences and online fora. It is seen in e.g. fibromyalgia, CFS/ME and PTSD. See e.g.... <https://www.psychiatryadvisor.com/home/topics/neurocognitive-disorders/alzheimers-disease-and-dementia/stress-increases-the-risk-of-mild-cognitive-impairment/> OR <https://www.verywellhealth.com/brain-fibro-fog-causes-symptoms-possible-treatment-716014> OR . It is totally unnecessary and misleading that the authors write as if we know little about this phenomenon from before. The background is in clear need of an update on the science of persistent symptoms and “brain fog” or e.g. cognitive impairment more generally.

We’ve addressed this point above, in particular in the initial theoretical framework discussion on page 11.

METHODS

- P 12, line 10. Although frequently used, the terms “physical symptom” is nonsense (in a philosophical sense of being conceptually inadequate). There are no symptoms that are not subjective or “psychological” per definition. I take it that the authors do not really mean that cognitive symptoms do not refer to something physical (the brain). If the authors are going to use such conceptual language, they should define what they mean by a “physical” symptom as opposed to a “neurocognitive” symptom. Please add in the definition list.

We use the term ‘physical’ to refer to a symptom which the participant described as physical. This is in keeping with our patient-centred study design in which we try to convey the patient’s lived experience. We have tried to define what participants meant by ‘physical’ on page 14 in the paragraph about ‘neurocognitive symptoms in the context of other long Covid symptoms’.

THERETICAL FRAMEWORK

- P 12, line 55. A definition of a neuroscience perspective is NOT to think that the cause needs to be SARS-COV2 disrupting the brain and the brain stem. Neuroscience can be said to be a reductive (and productive) endeavor trying to reduce e.g. symptoms to brain function, but such a perspective does NOT necessarily involve thinking that SARS-COV2 disrupts or is the cause of the problems in itself. This is a pretty serious theoretical mistake and statement. The reference to allostasis is a good one, neuroscience is not just about the brain. However, there is no theoretical divide between allostasis and homeostasis. Allostasis is about keeping homeostasis through dynamic change. It is faulty to state that systems of homeostasis or allostasis AND physiology INTERACT with brain systems of mood etc. Systems of mood, attention etc of course is PART OF the body’s allostatic systems. The allostatic theoretical perspective which seeks to overcome the divide between “psychological” and the body is largely missing in the text further on (McEwen).

We have amended the theoretical framework on page 11.

- P. 13. NB! I think the theoretical perspective is sorely lacking a reference to the science of persisting systems. I think the authors should be required to refer to these and refer to them. There is a large body of science trying to explain just what the authors are dealing – the bridge between the perspective of neuroscience and the perceived symptoms - with here and they don’t refer to it. I can’t see that as adequate. See again
<https://www.sciencedirect.com/science/article/abs/pii/S0272735820300179> or
https://core.ac.uk/reader/80805780?utm_source=linkout

We have amended the theoretical framework on page 11 and have included several references – as discussed above – which better accord with our neuroscience perspective.

RESULTS

- P. 14-15 lines 50 and onward: It may seem here to the reader as if the patients have been subjected to a neuropsychological examination. They have not. The authors do not know that their experiences would match “objectively” assessed cognitive function. This should be made clear. Their experiences would correspond to these domains and MAY correspond to such deficits but their actual lack of function in these domains have NOT been assessed.

We agree and have clarified this point on page 13.

- P 15, line 54: “Physical fatigue”. This is again an example of a concept that is nonsensical in a world without a mind-body dichotomy. What is “physical fatigue” exactly? What is the difference from “mental fatigue”? There is only one fatigue. If the authors think there is a difference and wish to use these messy terms they should define what they mean by them. The authors should explain what they mean by this or just use descriptive terms. What symptoms are we talking about?

We have amended the wording to say ‘a sensation of physical fatigue’. We respectfully disagree that there is only one fatigue. Some fatigue feels physical, some feels mental and some feels both.

- P16. The study lacks a reference to something very important: What does the “fog” mean in the particular individual’s life context. For example, someone working with something that demands a lot of cognitive attention or multitasking may feel differently than those who do not.

We agree this is an important point and have made it in the opening paragraph on page 15. The people complaining of brain fog often had jobs where they needed agile brains (e.g. a paediatrician who had to calculate drug doses for small children – a difficult and high-risk task)

- P 17, lines 18-30. The sense that one is “validated” if the brain fog is thought to be due to e.g. brain damage and that it is somehow non-valid that it is due to anxiety or depression or “in your head” is an important finding. The authors could elaborate that, do the subjects say more? However, the authors should watch their language and make sure that THEY don’t suggest and validate the idea that brain fog due to anxiety or depression is somehow second range or non-valid and thus contribute to the stigma against disorders like these (which likely underlies the respondents’ feelings of anger). The authors should not suggest that healthcare workers talking to patients about how anxiety or fatigue can cause brain fog is not validating them. This is disrespectful towards the sufferers of these conditions who suffer something every bit as real as anything else.

We accept this point and have checked our own language throughout (we’ve made one or two small changes to make sure we’re not inadvertently biased towards a ‘physical’ mechanism.

- P 17: Hypothesising mechanisms. The use of the term “mechanisms” is not neutral theoretically. By mechanism we often understand something on the molecular level or low biological level. What is hypothesized here is not mechanism, which the human experience cannot access through our senses, but causes more broadly. Using analogies of stroke for example is not about hypothesizing cause in a scientific sense, it is about the ideas these people have of their illness, their meaning-making. The theories they put forward are not taken from their own experiences, they are taken from something they have read about.

We agree this is an important point and have amended the language on page 15, changing ‘hypothesising mechanisms to inform self-management’ to ‘hypotheses to inform self-management’ and ensuring consistency of language in the associated paragraph.

-
DISCUSSION

- P 18: Discussion: The authors here claim to have revealed that the symptoms “included deficits”. They have not shown this. They have documented symptoms that MAY correspond to such deficits, but they have not performed any tests to check if these perceptions correspond to a deficit in these neurocognitive domains. This should be made perfectly clear.

We agree and have changed the wording on page 17.

- P. 18: line. Statements such as “interaction” between physical and cognitive symptoms are again philosophically nonsensical as there are no symptoms that are non-physical or non-cognitive. What do the authors actually mean? Please state this in clear language.

We accept this point and have changed the wording on page 17 to ‘interaction between perceptually cognitive or physical symptoms ‘ along with the afore-mentioned description of what patients mean by ‘physical’ symptoms on page 14.

- P 18, line 47: The authors take it as granted that the patients lacked neurocognitive skills. This is not something the authors know. This is an illness *perception* or *belief*. It can be, but is not *necessarily* so, that they perform very bad on tests.

We agree and have changed the wording on page 17 to ‘subjectively cognitively impaired’.

CONCLUSION:

- P 20, line 6-23. In the middle of what is an account of the experiences of people with the disorders the authors venture into an account of what ongoing research is supposed to show about pathophysiological cause. This small discussion is heavily skewed towards a reductive (and scary) view which completely omits both prior knowledge about such symptoms (which the authors should familiarize themselves with) and specific references showing links to e.g anxiety, see above. More importantly, this passage is out of place in a discussion about patient experience. They are frustrated with having no reductive pathophysiological explanation which in biomedicine denotes something “real”. The authors don’t have to help them with such a reductive explanation. It is not their project or research question in this article. This passage should be deleted. More should be written about this experience, which is important. Why is it important for them to have a specific explanation?

We have addressed this by including the paragraph on theoretical frameworks on page 11, which attempts to introduce a greater balance and have tried to introduce a more-open minded framework by changing the text on page 19 to read: ‘whatever the explanation for ongoing neurocognitive symptoms, the resultant impacts result from – and contribute to – a wider interplay of psychological, physical and social factors.’

- P 20: Results/Discussion: General: A strength is talking to people who have recovered. I would want more discussion about how the narratives of those who recover differ from those who do not. This is very important variation in the dataset which should be explored.

It was striking that the main difference between those who had recovered and those who hadn’t was the fact of recovery. The recovered people’s recollections of brain fog were very similar to the current experiences of those who had not recovered. We have clarified this in the paper on page 20.

- P 21 Selection: The important problem with selection of people who frequent online fora as representative is noted discussion on limitations. The authors should refer to research showing that these are the patients who experience the most distress.
<https://pubmed.ncbi.nlm.nih.gov/32758507/>

Thanks. Added on page 20.

- P 22: NB! Again, the discussions lacks connection to the science of persistent symptoms

This reviewer has a hypothesis about the science of persistent symptoms – but this is based on other diseases and we know of no evidence that the science is the same. We can't simply “connect to the science” when the link in this condition is assumed rather than proven. Accordingly, we've given it a mention but it would be highly unscientific to simply cut and paste this reviewer's strong opinions into the paper.

- P 22, line 38. The authors state that infection with a list of microbes RESULT in the impairments. This has not been shown. The authors need to distinguish between initiating causes and causes that cause chronification. The causes of persistent infections are an empirical question. *Borrelia burgdorferi* does not RESULT in all these symptoms. On the contrary, the infectious disease community have long struggled to communicate that the symptoms experienced after Lyme disease are probably caused by other factors (see e.g. Feder et al., 2007, NEJM).

We have changed the wording on page 21 to 'can be associated with' and added a rider that the causal link is disputed.

- P 23: line 28: Again, the depiction of the disabilities as “neurological” is misleading as it suggests organic damage.

Changed to “neurocognitive” which we've defined earlier in the paper.

- P 24: Conclusion. NB! Again, here the authors state something important about their objective: To align the subjective illness experience and objective disease models. This is connected to my most important criticism here: The lack of reference to and theoretical use of literature that already makes this connection and has studied it for a long time. See references above. Relatedly, it is a huge problem that the authors write as if brain fog is something specific or new to post-covid syndrome. It is not.

The reviewer does not know whether the brain fog of covid is new or not, though we can see he has a firm opinion on that matter. We acknowledge that we ourselves do not know if it is new or not, and we had implied it must be. In the revised version we've acknowledged that it COULD be new or it could have parallels to the fog of other diseases. Moreover, we have amended the text on page 22 to reflect a more balanced viewpoint that now reads, 'align the subjective illness experience with the perception of neurocognitive symptoms and proposed causal and contributory hypotheses.'

- P 24: I think the authors should refrain from the political argument above preventing covid. It is not connected to their research question. It could also be noted that it is ethically tenous for researchers to contribute to anxiety and symptom focus about a condition, as this will likely increase the very symptoms and distress they are investigating (see Barsky, JAMA, 2017: The iatrogenic potential of physician's words).

We don't think a statement about preventing Covid is political any more than saying it's a good idea to prevent diabetes. And that goes whether the sequelae are due to persistent viraemia or to anxiety or to anything else. It would be better to prevent Covid.

- P 24: “Neurotropic” virus. It is unjustified to present this virus in this way. It has no special affinity for nerve cells. It is a respiratory virus with systemic effects (like many other viruses). Stick with “virus”.

We disagree. The neurotrophic nature of SARS-CoV-2 is well established and we have included the relevant reference on page 11 in the discussion on theoretical framework.

- P 25, line 35-37: It is not supported that the patients have actually not been “believed” or “supported”. This is about their experiences and their agency in portraying things. I think, again, the authors should avoid supporting the notion that connecting “brain fog” to e.g. anxiety it not about believing people. It is derogatory towards these forms of illness.

We have amended the wording slightly on page 23. It now reads, ‘described by individuals when their accounts are—respectively— perceived as acknowledged or dismissed underscores the importance of the clinical relationship in which the patient is listened to, their experience believed, and supported — particularly in primary care’

Reviewer: 2

Dr. Helen Humphreys, Sheffield Hallam University, Sheffield Hallam University

Comments to the Author:

This paper addresses an important and interesting phenomenon using an appropriate methodology and a commendable sample size. Whilst the findings are interesting, I feel that some sections could be developed further to ensure that the paper adds to the wider understanding of brain fog. I hope my comments are helpful.

Abstract

The conclusions stated in the abstract are quite broad and could easily be mistaken for general recommendations about managing long Covid. Can these be more specifically linked to what the paper has highlighted about brain fog in particular?

Good point. We have amended the conclusion to make it more specific to brain fog eg: ‘for example incorporating (where necessary) assessment and rehabilitation from clinical psychologists and occupational therapists’ and ‘the varied nature of the severe impacts of neurocognitive symptoms identified in this study highlight the importance of ensuring that specialist services are accessible, easily navigable, comprehensive, and interdisciplinary’

Background

Could the authors add a brief, critical summary of previously published qualitative research on long Covid to strengthen the rationale for this in-depth exploration of brain fog?

We’ve added a summary of qualitative studies in the background section on pages 7 and 8, which we hope will help with this rationale.

Methods

Please clarify the consent process - were participants provided with written information about the study and the way their data would be used? How was consent obtained from each individual participant?

Added, page 10 and a reference to the original study where the consent process is discussed in additional detail. Unfortunately the word limit here prohibits an extensive discussion.

How many researchers were involved in the data collection and analysis and how were the tasks divided? Under “data management and analysis” the term “we” is used but it isn’t clear to what extent investigator triangulation or similar took place.

Added, page 10.

It's great to see the theoretical lens for the research outlined in detail but would be useful to have a brief line to describe how this informed the analysis.

Added, page 10.

Patient Involvement Statement

It would be useful to have an indication about the types of modifications that were made to the draft paper based on participants' feedback.

Added, page 12.

Results

Can a list of the final themes be added at the start of the results section? If possible, it would be easier to follow if the themes names were consistent across the abstract and the results section.

Added, page 13.

To be honest, I found it difficult to follow the results with the participant quotes provided in a separate table. I acknowledge this may be due to word count limitations and also because some quotes are referred to more than once but if it's possible to include them in the main text it would bring the themes to life much more (perhaps aim to provide different quotes rather than repeating some to make use of the "1000 pages of transcripts and notes" that were collected).

Sorry, this was a word count issue but in the final printed version we think it will be easier to follow.

Whilst the themes were relatively self-explanatory, I often found myself wanting to learn a little more. For example, Neurocognitive symptoms - was it common for participants to experience symptoms across different domains of cognitive function? A key point highlighted in other long Covid 'lived experience' research has been the interplay between physical and cognitive symptoms and the associated impact on managing fatigue and activity. Did the authors learn anything further about how participants felt that these elements of their condition affected one another?

We would have loved to discuss this in more depth, but sadly were constrained by the word limit. We hope the section on page 14 about 'neurocognitive symptoms in the context of other symptoms' goes some way towards explaining this.

I found 9 themes a lot to read. I don't wish to dictate what themes the authors choose to present, but I do wonder if a little more development/analysis might allow some of the themes to be combined into something more comprehensive - e.g. Psychosocial impact of neurocognitive symptoms / Guilt, shame and stigma; Hypothesising mechanisms / Self-management

We agree, and have combined some of the themes so there are now only six.

Navigating the healthcare system - the challenges described here echo previous lived experience research and these have been reasonably well documented to date. The particular challenges of communicating and self-advocating whilst experiencing brain fog are perhaps not emphasised

enough. Were there any particular neurocognitive symptoms that were particularly likely to be dismissed by healthcare professionals? What sort of investigations had healthcare professionals been willing to offer or refer to? What was the role of long Covid clinics in this experience (or were these not yet established)?

It's difficult to break this down in a detailed way to represent a consistent picture as experiences varied so widely, although we have mentioned it on page 16, and due to the timing of the initial recruitment, during the first wave of the pandemic when long Covid services were not even thought of, let alone established, the vast majority had not seen any specialists. We agree that it would be an interesting element to explore in further studies.

Discussion

The opening paragraphs of the discussion section paint a compelling picture of how challenging the experience of brain fog can be which I am not sure has come through in the results section. We've amended the results section to incorporate this.

The discussion could be strengthened with more specific discussion about the key points raised by participants. For example, the term "brain fog" was not considered to accurately describe some peoples' experience. Is it appropriate to keep using this or might another term be more useful?

Most people liked the term brain fog actually, though one or two didn't. Moreover, despite not liking it – specifically because they felt it downplayed the 'physical mechanisms' they believed underpinned their symptoms – even those few individuals were content to use the term in a conversational context as they felt it did adequately capture their experience. We have tried to convey this in the results on page 13.

What does this paper tell us about how to strike a balance between medical investigation of neurocognitive symptoms that helps to understand the causes and mechanisms against the application of self-help strategies? Is brain fog an inevitable consequence of long Covid that people with the condition will have to simply endure or are there specific things that either they, or people supporting them can do to make it more bearable and/or alleviate the challenges somewhat? How are neuropsychiatric symptoms currently being addressed (and by whom) within long Covid rehab pathways and what changes/recommendations to those pathways does this study suggest?

We have tried to incorporate some of these issues in the discussion on theoretical frameworks on page 11 and the Conclusion section. We would argue that Covid rehab pathways are currently highly heterogeneous and determined by local resource constraints, therefore the best we can suggest – as we do here – are guidelines that advocate for a multidisciplinary, easily-accessible approach that includes psychologists, cognitive neurologists, and occupational therapists (page 23). We recognize that some already do this, but, based on the impact reported in this paper, we feel this is worth emphasizing again.

The impact of these symptoms on peoples' ability to return to work or maintain their regular employment schedules is not insignificant. Is there anything to be said about the types of roles/jobs/employees that are particularly impacted by this? Besides reducing hours, what reasonable adjustments could employers be making?

Sadly we are limited by word limit in exploring this in depth and this study did not directly explore occupational changes that might be helpful. However, on page 23, we have argued for further exploration of processes to aid occupational restoration as we completely agree that this is crucial.

Limitations

The absence of people who are digitally excluded appears to be a recurring limitation in long Covid research - can the authors make any comment or recommendation about how future research might overcome this? It might also be prudent to acknowledge this in the strengths and limitations section at the start of the paper.

Good point. We've commented on this on page 20. Again we would love to comment further about research approaches to optimise inclusion of digitally-less connected groups. However, the word count limit prevents our being able to do so here – it is beyond the scope of this paper.

The full paper requires a thorough proof-read and check for spelling and grammar.

Agree, done!

Reviewer: 1

Competing interests of Reviewer: I am an MD and postdoctoral researcher in medical scientific theory at the Centre for medical ethics in Oslo. I'm also the leader of Recovery Norway, an organization consisting of people who have recovered from disorder such as ME/CFS and are similar to post-covid syndrome through strategies involving changes in e.g. behaviour and cognition. As such I have an intellectual conflict of interest when it comes to how such disorders may be mitigated.

Reviewer: 2

Competing interests of Reviewer: None

1. Perego E, Callard F, Stras L, Melville-JÛhannesson B, Pope R, Alwan N. Why the Patient-Made Term 'Long Covid' is needed. Wellcome Open Research. 2020;5:224.

VERSION 2 – REVIEW

REVIEWER	Humphreys, Helen Sheffield Hallam University, Advanced Wellbeing Research Centre
REVIEW RETURNED	14-Dec-2021

GENERAL COMMENTS	Thanks to the authors for their responses. I welcome the changes made to this manuscript which provide increased clarity on the findings. The paper now better addresses the research question/aims, with more relevant discussion and conclusions as a result. I support the authors' stance on retaining patient-defined terms e.g. brain fog, long Covid in the spirit of patient-centred research.
--